# A Comparative Multianalytical Approach to the Characterization of Different Grades of Matcha Tea (*Camellia sinensis* (L.) Kuntze)

**DOI:** 10.3390/plants14111631

**Published:** 2025-05-27

**Authors:** Chiara Toniolo, Adriano Patriarca, Daniela De Vita, Luca Santi, Fabio Sciubba

**Affiliations:** 1Department of Environmental Biology, Sapienza University of Rome, Piazzale Aldo Moro 5, 00185 Rome, Italy; chiara.toniolo@uniroma1.it (C.T.); daniela.devita@uniroma1.it (D.D.V.); fabio.sciubba@uniroma1.it (F.S.); 2Department of Chemistry, University of Rome Sapienza, Piazzale Aldo Moro 5, 00185 Rome, Italy; adriano.patriarca@uniroma1.it; 3NMR-Based Metabolomics Laboratory (NMLab), Sapienza University of Rome, Piazzale Aldo Moro 5, 00185 Rome, Italy; 4Interdepartmental Center of Applied Sciences for the Protection of the Environment and Cultural Heritage (CIABC), Sapienza University of Rome, Piazzale Aldo Moro 5, 00185 Rome, Italy

**Keywords:** matcha tea, catechins, amino acids, polyphenols, theanine, HPTLC, NMR

## Abstract

Matcha, a finely powdered green tea, has been cherished in Japan for centuries, used in the traditional tea ceremony and nowadays also valued for its health-promoting properties. Cultivated under shaded conditions to enhance chlorophyll production, which gives the typical vibrant green color, matcha is rich in important bioactive compounds, including caffeine, catechins, and theanine. This study analyzes three matcha grades—ceremonial grade 1 (G1), grade 4 (G4), and food grade (FG)—to assess variations in their metabolite profiles. The Bligh–Dyer method was employed to extract polar and non-polar metabolites from organic and hydroalcoholic phases. High-performance thin-layer chromatography (HPTLC) was used for qualitative metabolite analysis, while nuclear magnetic resonance (NMR) spectroscopy was employed for both qualitative and quantitative analyses. Results reveal a decreasing gradient of amino acids and caffeine from grade 1 to food grade, while other metabolites, such as polyphenols, display an increasing trend. These findings suggest that factors such as harvesting time and leaf maturity significantly influence matcha’s chemical composition, providing a scientific basis for its quality differentiation and potential nutraceutical uses.

## 1. Introduction

Tea is a well-known and widely consumed beverage worldwide, made by infusing leaves of *Camellia sinensis* (L.) Kuntze in water. Various types of tea are classified based on the degree of oxidation and the processing methods applied to the leaves [1,2,3,4]. Matcha is a finely powdered green tea regarded as a premium product. Used for centuries in Japan, it has traditionally been consumed by Buddhist monks as a healthful drink for both the body and mind.

Matcha cultivation occurs under shaded conditions, traditionally using bamboo mats shielding the leaves from direct sunlight. This technique boosts chlorophyll production, giving matcha its characteristic bright green color [4,5,6]. Matcha tea is commercially classified into different grades: ceremonial grade, the highest quality, is used for traditional tea ceremonies and has a smooth, delicate taste, while the food grade is slightly more bitter and intended for cooking. The harvest, for higher grades conducted exclusively by hand in early May, involves selecting young tea buds and leaves. These are subsequently steamed to prevent natural oxidation, then dried and cooled. The leaves are then trimmed to remove stems, stalks, and older leaves. Finally, the tea is ground with handcrafted stone mills to produce a fine powder [7,8,9].

Unlike other types of tea, matcha is consumed as a powder mixed with hot water. Its traditional preparation involves a large ceramic bowl (*cha-wan*), and a hand-carved bamboo whisk (*cha-sen*) used to create the distinctive “jade foam”. Thus, matcha consumption facilitates the ingestion of the entire spectrum of bioactive compounds found in tea leaves, not just those extracted in a traditional infusion [1,4].

The quality and final grade of matcha are determined by several factors, each contributing to the tea’s unique characteristics and intended use. Key determinants include the type and duration of shading, the timing of harvest, leaf size, precision in removing stems and veins, the speed of grinding, and the type of mill used [1,4,7].

Shading plays a crucial role in the flavor and nutrient content of matcha. In particular, the specific cultivation techniques—especially shading—as well as the processing methods, lead to an increase in amino acids, caffeine, and chlorophyll content, and a decrease in catechin content. These compounds, besides being bioactive constituents, significantly influence the organoleptic characteristics of matcha [4,5,7,10,11]. High-quality ceremonial grade matcha is typically harvested in spring, when the leaves are younger and more tender. In contrast, food-grade matcha is often harvested later, during the summer [7,12,13].

The size and age of the leaves are other defining factors. Grade 1 ceremonial matcha is obtained exclusively from buds and first leaves, representing the finest and most delicate selection. Other ceremonial grades are typically made using the second and third leaves, which, while still tender and high in quality, offer a slightly different flavor and texture profile. Lower grades (e.g., grade 4 and beyond) incorporate larger, older leaves, such as the fourth or fifth leaf, resulting in less delicate flavors and textures [5,14,15,16].

Meticulous attention is given to the removal of stems and veins in premium matcha, ensuring a smooth and silky texture. Additionally, the grinding process is critical: ceremonial grade matcha is traditionally ground using stone mills, a slow process that preserves the vibrant color and delicate flavor. Faster grinding methods or alternative milling techniques are typically used for lower grades, which have a more robust flavor and are therefore, better suited for food applications [5,7,8].

Ceremonial grade matcha is reserved for traditional tea ceremonies due to its superior quality, with grade 1 representing the pinnacle of craftsmanship and refinement. Conversely, lower-grade matcha, while less delicate, is ideal for recipes, that have been increasingly gaining recognition in recent times, such as ice creams, cookies, and smoothies, where its bolder and richer flavor can complement other ingredients [7,9,12,13].

The traditional cultivation of matcha promotes the synthesis of bioactive compounds such as theanine, caffeine, chlorophyll, and catechins, with epigallocatechin gallate being the most prevalent and potent [5,15,17]. This unique composition makes matcha a rich source of antioxidants and anti-inflammatory agents, offering notable physical and mental health benefits, including disease prevention and enhanced cognitive function [15,16]. Its bioactive profile, comprising L-theanine, caffeine, and phenolic compounds like quercetin, rutin, and catechins, effectively counteracts oxidative stress caused by an imbalance of reactive oxygen species (ROS), thereby supporting cellular health [15,16]. Other significant activities include anti-inflammatory effects, cardioprotective properties, and the potential to regulate carbohydrate metabolism [15,18].

Regardless of the grade, matcha tea is a premium product often considered exclusive and expensive. Understanding the qualitative and quantitative differences between various grades is essential not only for accurately evaluating product quality but also for enhancing consumer awareness and guiding its proper use. Furthermore, analyzing how metabolite levels change with leaf maturity offers valuable insights into the biological properties of matcha, potentially serving as indicators of its functionality, nutritional value, and health benefits. This research contributes to the nutraceutical field and food science in general by clarifying the chemical and qualitative factors that influence matcha tea grading and applications. Indeed, green tea's chemical profile has already been investigated as a function of quality [19], fermentation [20], pedoclimatic conditions [21,22], and cultivar [23]; yet, regarding matcha tea, while there are some studies [14,15,24], at the best of our knowledge, there is not a systematic investigation regarding how the matcha tea quality is related to its chemical profile.

Therefore, this study examines three different grades of matcha tea: ceremonial grade (grade 1—G1), grade 4 (G4), and food grade (FG). The samples are compared using very different analytical approaches to determine how the chemical composition varies based on the grade. The qualitative and quantitative determination of the constituents is performed using nuclear magnetic resonance (NMR) spectroscopy, while a chromatographic technique, specifically high-performance thin-layer chromatography (HPTLC), has been used for qualitative analysis. The choice of HPTLC and NMR for this study is based on their well-established reliability in food analysis. These techniques are not only rapid and efficient but also highly complementary. HPTLC is a cost-effective technique that offers excellent sensitivity, higher than NMR in the detection of certain compounds [25]. However, it is a targeted technique, whereas NMR can be applied in an untargeted metabolomic approach to provide a comprehensive profile of metabolites [26,27,28]. Both techniques are highly sensitive and, when used together, provide a robust and detailed analysis of the sample, offering a broad detection of the metabolic constituents, thus providing useful information about the composition and quality of food products such as matcha tea.

This study represents a preliminary investigation with multiple objectives, including the identification of potential marker compounds that could serve as indicators of quality and the development of a rapid and cost-effective method for their determination. This approach aims to provide a solid scientific basis for quality differentiation and the exploration of the potential nutraceutical applications of tea. Furthermore, the proposed methodology is intended not only for tea analysis but also for possible application to other plant matrices, dietary supplements, or related products.

## 2. Results and Discussion

The analyses carried out enabled the investigation of variations in the content of amino acids, caffeine, organic acids, phenolic compounds, and other miscellaneous molecules in matcha tea samples. Specifically, three different matcha tea samples were analyzed, belonging to three different quality categories, ranging from ceremonial grade to food grade.

The study was based on the combination of two complementary analytical techniques: HPTLC and NMR spectroscopy. The combination of these methods enabled a comprehensive assessment of the chemical composition, leveraging the sensitivity of HPTLC and the detailed structural insights provided by NMR. This dual approach ensures robust and accurate profiling of the bioactive compounds, offering valuable insights into the quality and functionality of matcha tea across different grades. The combined use of these two techniques has been reported for the analysis of a few specific compounds in C. sinensis leaves [17], but to the best of our knowledge, it has never been applied to matcha tea or to such a large number of metabolites. HPTLC is an extensively used chromatographic technique in food analysis, which has been specifically applied in the study of tea [29,30]. It enables the simultaneous analysis of extracts and standards, verifying the presence of the latter within the sample. The CAMAG TLC Scanner 4 measures the absorbance and fluorescence of compounds, also providing reliable quantitative data through densitometric peak profiles for individual tracks. The instrument operates within a spectral range of 190–900 nm and records the UV-VIS spectra of the detected peaks, allowing for a comprehensive evaluation of the data [31]. Unlike HPTLC, nuclear magnetic resonance (NMR) is a spectroscopic method based on the magnetic properties of the nuclei of certain atoms and isotopes and, therefore, does not require a direct comparison with standards. The identification of different substances within an extract is based on the interpretation of information related to the resonance frequency of active atoms’ nuclei.

The HPTLC analysis made it possible to verify the presence of various metabolites within the samples and to evaluate how their individual concentration varied from G1 to FG.

Comparing the NMR spectra of the three categories, both qualitative and quantitative differences were observed. From the examination of hydroalcoholic and chloroform phase spectra, a representative G1 assigned ^1^H spectrum is reported in Appendix A. Overall, 44 metabolites were identified and quantified, and their chemical shift, resonance assignment, and signal multiplicity from the ^1^H spectrum are reported in Appendix A. The molecule amount was calculated by comparing the integral of the diagnostic resonance of each molecule and compared to the integral of the internal reference, both previously normalized by the number of hydrogens. Then, the amount was reported as mg of the molecule for 100 mg of starting raw material.

### 2.1. Amino Acids Analysis

The HPTLC analysis from the hydroalcoholic phase of matcha tea grades G1, G4, and FG (Figure 1) examined a wide range of amino acids, including not only theanine but also several proteinogenic amino acids identified in previous studies [5,32,33]. Aromatic amino acids were also analyzed since, given the abundance of molecules containing aromatic moieties, their quantification through NMR is hindered by resonance superimposition. The amino acids analyzed include alanine, arginine, asparagine, aspartic acid, cysteine, glutamic acid, glutamine, glycine, histidine, isoleucine, leucine, lysine, methionine, phenylalanine, proline, serine, theanine, threonine, tryptophan, tyrosine, and valine.

Among the essential amino acids, lysine was clearly identified, whereas isoleucine and leucine could not be reliably distinguished. Using the mobile phase 1-butanol/acetone/acetic acid/water (7:7:2:4 *v/v*), these two amino acids show very similar R*f* values (0.68 and 0.69), as does tyrosine (Rf 0.69), making it difficult to determine whether they are present individually or as a mixture. In the samples, a broad spot is observed in this Rf range. A similar pattern was also observed for alanine and threonine, which show closely spaced R*f* values (0.42 and 0.43).

Preliminary analyses using an alternative mobile phase (1-butanol/acetic acid/formic acid/water, 28:9:8:2 *v/v*) excluded the presence of tyrosine but did not improve the distinction between leucine, isoleucine, and other amino acids.

Therefore, the mobile phase 1-butanol/acetone/acetic acid/water (7:7:2:4 *v/v*) is considered the most effective for identifying the largest number of amino acids, although it presents some limitations in separating compounds with very similar R*f* values.

This analysis, in addition to enabling the identification of amino acids, allowed us to observe a common decreasing trend of concentration from G1 to FG of all detected amino acids. In fact, G1 appears to be particularly rich in amino acids, with theanine being the most abundant compound. This decreasing trend maintains the concentration ratios of the different substances, indicating that the reduction occurs proportionally across the different tea samples.

Preliminary analyses have excluded the presence of asparagine, cysteine, glycine, histidine, methionine, phenylalanine, proline, tryptophan, and valine.

The amino acid contents evaluated by densitometric scanning analysis of HPTLC plates are compared in Table 1 while the quantification by NMR spectroscopy is reported in Table 2.

Densitometric scanning data are available in the Appendix A.

NMR analysis allowed the identification of the following amino acids: alanine, dimethylglycine, glutamine, isoleucine, leucine, lysine, theanine, threonine, and valine.

Dimethylglycine (DMG) is a naturally occurring derivative of glycine, characterized by the presence of two methyl groups attached to the nitrogen atom. As an *N*,*N*-dimethylated amino acid, it plays a role as an intermediate in the metabolic conversion of choline to glycine betaine, a pathway often linked to cellular stress responses. In plants, DMG is typically associated with osmoprotective functions and methylation cycles. Although not widely studied, its presence has been reported in several species [34,35], including *C. sinensis* [36], where it may contribute to adaptive responses and secondary metabolism. Univariate analysis showed significant differences among the amino acids between G1 and G4 versus FG, with generally higher levels observed in G1 and G4, in general agreement with HPTLC data. In particular, higher amounts of glutamine, isoleucine, leucine, and lysine were reported in G1.

The HPTLC analysis of amino acids in the hydroalcoholic extracts of matcha tea G1, G4, and FG allowed the definitive identification of theanine, along with proteinogenic amino acids: arginine, aspartic acid, glutamine, glutamic acid, and serine. Additionally, several essential amino acids were identified, including lysine, isoleucine, and leucine.

Aromatic amino acids were also tested but were not identified in the samples. This test was performed because, in the corresponding spectral region of NMR experiments, the most abundant resonances belonged to the phenols and polyphenols, thus making a necessity to employ a complementary targeted approach.

As previously noted, theanine—a γ-amino acid—was the most abundant amino acid, especially in sample G1. Known for its health benefits, such as promoting relaxation, enhancing learning ability, and boosting the immune system [22], theanine played a key role in distinguishing tea quality. The analysis revealed a clear trend: amino acid concentrations decreased progressively from G1 to FG grade, with G1 standing out for its richness in amino acids, particularly theanine. This pattern aligns with existing literature [5].

The most plausible explanation lies in leaf aging. The observed general decrease in amino acid content from younger to older leaves can be attributed to the higher metabolic activity and protein synthesis in young tissues, combined with the remobilization of nitrogenous compounds from senescing leaves. This dynamic reflects the plant’s strategy to optimize nitrogen use and support growth and development in metabolically active sink tissues.

This phenomenon is particularly evident in the case of theanine, the most abundant non-proteinogenic amino acid in *C. sinensis*, which plays a central role in nitrogen storage and transport. Synthesized primarily in the roots, theanine is translocated to young leaves, where it contributes to nitrogen buffering and supports active protein biosynthesis. As leaves mature and enter senescence, theanine, along with other nitrogenous compounds, is remobilized toward younger organs, thereby explaining its lower accumulation in older leaves [37].

Consequently, higher-quality teas tend to contain higher levels of amino acids. In fact, according to the literature, premium green teas, including matcha, are particularly rich in amino acids. Notably, Hideki Hori et al. (2017) [5] reported that total amino acid content may serve as a potential indicator of green tea quality.

### 2.2. Caffeine Analysis

To determine caffeine, an additional HPTLC analysis was performed. Although the mobile phase used for catechins determination allowed clear visualization of caffeine, catechins became distinctly visible only after derivatization (see Section 2.3). However, catechin, epicatechin, and caffeine share the same R*f*; additionally, catechins absorb slightly at 254 nm, though to a lesser extent than caffeine.

To ensure accurate data interpretation and prevent spot overlap a specific analysis for caffeine was conducted, ensuring distinct and non-overlapping R*f* value (Figure 2, caffeine R*f* 0.45).

The analysis of caffeine, a xanthine alkaloid, as expected, shows that it is particularly abundant in the organic phase, while also present in lower amounts in hydroalcoholic extracts. However, unlike polyphenols, caffeine exhibits an opposite trend: grade G1 has a higher caffeine content compared to FG (Figure 2), following the same pattern observed for amino acids.

Additional analyses (data reported in the Appendix A) investigated the presence of the other two xanthine alkaloids: theobromine and theophylline. These compounds were not detected in the samples, likely due to their low concentrations or to limitations in the extraction method, which may not have effectively or fully extracted them. It is well known that these compounds are present in tea, although at lower concentrations than caffeine, which, as previously mentioned, is particularly soluble in chloroform.

NMR analysis also shows a decreasing trend in caffeine concentration from G1 to food grade (FG). Specifically, caffeine concentrations (mg/100 g) were found to be 472.53 ± 17.76 for G1, 386.6 ± 30.49 for G4, and 260.28 ± 4.24 for FG.

The organic extracts, as expected, showed a higher caffeine content compared to the hydroalcoholic extracts. In general, this secondary metabolite was found to be highly abundant and exhibited a concentration trend opposite to that of other secondary metabolites: specifically, G1 had a higher caffeine content than FG. This finding is supported by literature data. In fact, it is well known that the caffeine content in tea leaves varies depending on several factors, including the cultivar, leaf developmental stage, environmental conditions, harvest season, and degree of oxidation. In general, young leaves contain more caffeine than mature ones, with differences reaching up to 40% [38].

This pattern can be explained by the biological role of caffeine in the plant. Caffeine acts as a chemical defense compound, deterring phytophagous insects. Young leaves, being more tender and vulnerable, are more reliant on such protective mechanisms. As a result, caffeine biosynthesis is more active in the early stages of leaf development, leading to its greater accumulation in younger tissues. In contrast, as leaves mature and lignify, their dependence on chemical defense lessens, and caffeine levels tend to decline accordingly [39].

### 2.3. Organic Acids and Phenols

A single analysis was performed for both organic acids and flavonoids, as the selected mobile phase, chosen based on previous experience, enabled clear separation and visualization of both chemical classes under identical conditions, without any overlap issues (Figure 3).

The analyzed compounds include the following: 3,5-di-caffeoylquinic acid, apigenin, caffeic acid, chlorogenic acid, cinnamic acid, gallic acid, hyperoside, kaempferol, luteolin, luteolin 7-*O*-glucoside, protocatechuic acid, quercetin, rutin, and shikimic acid.

Among the flavonoids, the presence of rutin and kaempferol was identified, with a concentration increasing from G1 to FG.

Conversely, among the organic acids, only chlorogenic acid and shikimic acid were identified, showing an opposite trend compared to flavonoids.

In this analysis, as well as in preliminary analyses, none of the following compounds were detected: 3,5-di-caffeoylquinic acid, apigenin, caffeic acid, cinnamic acid, gallic acid, hyperoside, luteolin, luteolin 7-*O*-glucoside, protocatechuic acid, and quercetin.

Flavonoids and organic acids contents are compared in Table 3.

Densitometric scanning data are available in the Appendix A.

In other studies, some of these compounds were detected [40,41]. This observed discrepancy can likely be attributed to variations in the analytical conditions, with a particular emphasis on the extraction methodology employed. Differences in metabolite recovery and composition are known to arise based on the choice of extraction protocol, solvent polarity, and sample preparation techniques. In our case, we utilized the Bligh–Dyer method, a widely recognized biphasic extraction technique that employs short volumes of a chloroform–methanol–water system, which could be suboptimal for low solubility compounds. Another possible major factor could be the tea cultivar as assessed by Monobe et al. [40]. Indeed, in this work, the concentrations of tea polyphenols were very heavily dependent on plant cultivar and pedoclimatic conditions.

Concurrently, NMR analysis allowed the identification of the following organic acids: 4-hydroxybenzoic acid, cinnamic acid quinic ester, protocatechuic acid, gallic acid quinic ester, total chlorogenic acids, and quinic acid. The corresponding concentrations are reported in Table 4.

Univariate analysis showed, among organic acids metabolites, a decrease in concentration from G1 to G4 and FG of gallic acid quinic ester, quinic acid, and protocatechuic acid (which has not been detected in G4 and FG), where the total content of chlorogenic acids and cinnamic acid quinic ester is higher in FG and 4-hydroxybenzoic acid in G4.

Flavonoid and organic acid profiling revealed the presence of kaempferol, rutin, chlorogenic acid, and shikimic acid. Notably, rutin and kaempferol were most abundant in FG, with their concentrations progressively decreasing toward G1. Conversely, organic acids, particularly shikimic acid, exhibited an opposite trend, increasing from FG to G1.

This inverse correlation aligns with the known role of shikimic acid as a key intermediate in the shikimate pathway, which produces phenolic compounds such as flavonoids, and other secondary metabolites important for plant defense and pigmentation. The higher accumulation of shikimic acid in G1 likely indicates a slower conversion into flavonoids, leading to lower levels of final products. In contrast, the elevated levels of rutin and kaempferol in FG suggest a more active flavonoid biosynthesis, possibly due to increased enzyme activity or gene expression in the earlier stage. Overall, these results highlight a coordinated relationship between precursor levels and the production of secondary metabolites during plant development [25].

HPTLC analyses did not detect the presence of protocatechuic acid, whereas NMR spectra showed peaks that could be assigned to this compound. Since the sensitivity of HPTLC is higher than that of NMR and direct comparison with standards yielded negative results, it is possible for this molecule to be an ester or a glycoside rather than its free form.

A separate analysis was conducted for the study of catechins. The HPTLC analysis also allowed the identification of catechin and its derivatives. The catechins analyzed were as follows: catechin, epicatechin, catechin gallate, epicatechin gallate, epigallocatechin, and epigallocatechin gallate.

All the analyzed compounds were found to be present in abundant concentrations. Additionally, as clearly shown in Figure 4, a gradual increase in these secondary metabolites is evident, following an upward trend from G1 to FG. It can be observed that among these metabolites, epigallocatechin is the most abundant (R*f* 0.46), followed by epigallocatechin gallate (R*f* 0.41), epicatechin (R*f* 0.55), and finally epicatechin gallate (R*f* 0.51).

Catechin gallate is the only compound not detected in the samples, while catechin and epicatechin exhibit very similar R*f* values. Therefore, as previously noted for the amino acids leucine and isoleucine, it is difficult to determine whether both compounds are present or only one of them.

Densitometric scanning data are available in the Appendix A.

NMR analysis allowed the identification of EC, ECG, EGC, and EGCG (Table 5). Univariate analysis of these compounds showed that EC and EGC levels are lower in G1 compared to G4 and FG, while an opposite trend was observed for the gallate derivatives ECG and EGCG.

The catechins confidently identified were ECG, EGC, and EGCG. Due to overlapping R*f* values, catechin and epicatechin could not be reliably distinguished, and catechin was not quantified via NMR.

Despite these limitations, our results showed that the total catechin content was remarkably high, ranging between 5 and 6 g per 100 g of matrix. This is consistent with the well-established fact that green tea contains higher levels of catechins compared to other types of tea, such as black tea [15].

These findings agree with other studies; in fact, other works have shown that high-grade teas tend to contain lower levels of total catechins compared to lower-grade ones. Specifically, Horie et al. (2017) observed that EC and EGC contents were higher in lower-grade teas, while no clear correlation was found between tea grade and EGCG or ECG levels [5].

The higher catechin content in lower-grade teas is typically associated with the use of older, more senescent leaves, which accumulate greater amounts of these metabolites. Conversely, younger leaves, used in premium-grade teas, naturally contain lower levels of EC and EGC. Furthermore, shading treatments, which are common in the cultivation of matcha, are known to reduce the concentration of EGC and EC, contributing to the generally lower catechin levels found in high-grade products.

About food-grade matcha, drawing clear conclusions is more challenging, as these products are often blends of powders differing in leaf age, harvest time, and shading conditions. As a result, the biological variability of the samples significantly hinders the identification of consistent trends [5,42].

In addition to cultivation method, leaf age, and tea type, it is important to consider other factors that may influence the polyphenol content, such as catechins. Among these, brewing time and temperature are known to play a significant role in modulating the extractability and final concentration of these compounds [15,16].

### 2.4. Other NMR Detected Molecules

Quantified molecules were found belonging to different classes, such as polyols and carbohydrates (fructose, fucose, myo-inositol, sucrose, trehalose), lipids and sterols (cholestanol, ergostenol and ergosterol (quantified together), glycerophospholipids, FA-omega 3, FA-omega 6, FA-omega 9 (fatty acids), triacylglycerols, other metabolites (betaine, choline, α- and β-farnesene, methylguanidine, pyropheophorbide A, pyropheophorbide B, pheophytin A, pheophytin B), and also several different unknown compounds (U01 (myo-inositol glycosilate a), U02 (myo inositol glycosilate b), U03 (caffeoyl quinic acid), U04 (uridine)). Quantification and significant differences for each metabolite and sample group are reported in Table 6.

Only for trehalose univariate analysis are reported differences between all three categories of tea, with a decrease from G1 to FG, whereas for myo-inositol and fructose, a decrease is present comparing G1 and G4 to FG. Also, among carbohydrates, fucose is observed only in G1 and G4 samples. The sterols cholestanol, ergostenol, and ergosterol also diminish from G1 to G4 and FG, whereas for the cholestanol, no observable signals are present in the last group of samples. A different trend is observed for FA, since a decrease is reported for FA-omega 3 and 6, while for omega 9 higher levels are reported for the FG, followed by G1 and G4. Higher quantities of glycerophospholipids are present in G1, triacylglycerols are instead higher in FG. Choline, α and β farnesene, and methylguanidine are all decreased in FG compared to G1 and G4. For all pheophytins and pyropheophorbide, a reduction in quantities from G1 to G4 and FG is observable. In particular, the detection of pyropheophorbide A and pyropheophorbide B in processed *C. sinensis* leaves is plausibly related to the thermal treatment applied during tea production, particularly aimed at the inactivation of polyphenol oxidase (PPO). Pyropheophorbides are known to arise from chlorophyll breakdown, via pheophytin and pheophorbide intermediates, under heat or acidic conditions [43]. Their presence is therefore consistent with the sequence of biochemical transformations expected during leaf steaming or pan-firing, which not only deactivates PPO but also alters pigment profiles. This suggests that pyropheophorbide A and B are likely secondary products of chlorophyll degradation, formed as a consequence of the specific processing methods employed to preserve the quality and chemical composition of tea. Among the unknown compounds, U01 and U02 increase from G1 to G4 and FG, while G4 diminishes from the same set of samples.

## 3. Materials and Methods

### 3.1. Chemicals and Reagents

Standards and solvents were purchased from Sigma (Sigma-Aldrich, Milano, Italy). All chemicals and solvents were analytical grade. A complete list of all chemical reagents and standards used is provided in a dedicated paragraph in the Appendix A.

The stationary phase was HPTLC plates precoated with silica gel 60 F254 (20 × 10 cm) purchased from Merck (Merck, Darmstadt, Germany).

### 3.2. Samples Collection and Extract Preparation

Three different commercial matcha tea products from the Uji region in Japan were purchased: G1 and G4 were obtained from the seller Matcha Uji BIO (lot No. G-40621 and lot no. G-10018, respectively), while FG was sourced from Mondo Matcha (lot no. ZN21-21). G1 is a ceremonial grade made exclusively from the first buds, whereas G4 and FG include larger, older leaves, such as the 4th or 5th leaf. The samples were extracted using the Bligh–Dyer method: 250 mg of each sample were treated with 2 mL of CHCl_3_, 2 mL of MeOH, and 1 mL of H_2_O. The mixtures were vortexed and left to rest for 24 hours at 4 °C [44]. Subsequently, the samples were centrifuged, and phases were collected. The separation of the organic and hydroalcoholic phases yielded a total of six fractions: G1 organic fraction, G1 hydroalcoholic fraction, G4 organic fraction, G4 hydroalcoholic fraction, FG organic fraction, and FG hydroalcoholic fraction. All solutions were evaporated to dryness under a nitrogen flow.

For HPTLC analysis, an aliquot of each extract was diluted with the appropriate solvent for each phase (CH_2_Cl_2_ for the organic fraction and MeOH/H_2_O for the hydroalcoholic fraction) to achieve a final concentration of 5 mg/mL. Chemical reference substances were individually dissolved in methanol to obtain a concentration of 1 mg/mL.

### 3.3. HPTLC Analysis

Samples and standards were applied to HPTLC Silica Gel 60 F254 20 × 10 cm plates (Merck, Darmstadt, Germany) using the automatic sampler (ATS 4 Camag, Muttenz, Switzerland). The extracts were applied by an automated “spray-on” technique under an N_2_ flow at a rate of 100 nL/s, with a volume of 6 µL per application.

Plate development was carried out in the Automatic Developing Chamber 2 (ADC2—Camag, Muttenz, Switzerland) using the selected mobile phase for both development and saturation. Different mobile phases were employed depending on the chemical class to be analyzed: for flavonoids, ethyl acetate/dichloromethane/acetic acid/formic acid/water (100:25:10:10:11, *v*/*v*); for polyphenols, toluene/acetone/formic acid (9:9:2, *v*/*v*); for alkaloids, ethyl acetate/methanol/water (20:2.7:2 *v*/*v*); and for amino acids, 1-butanol/acetone/acetic acid/water (7:7:2:4, *v*/*v*). The choice of developmental solvents (i.e., solvent type and ratios) was guided by literature references and prior expertise [29,45].

After development, the plates were dried for 5 minutes at 120 °C on a TLC Plate Heater III (Camag, Switzerland) and visualized using the TLC Visualizer (Camag, Muttenz, Switzerland) at three different wavelengths: UV 254 nm, UV 366 nm, and white light.

For densitometric analysis, the TLC Scanner 4 (Camag, Muttenz, Switzerland) was used, with the detection wavelength set to different values depending on the metabolites: caffeine and polyphenols were detected at 272 nm, and flavonoids and organic acids at 272 and 330 nm, respectively, while amino acids were detected at 480 nm after derivatization with ninhydrin. The radiation sources were deuterium and tungsten lamps. The slit size was maintained at 5.00 × 0.20 mm, and the scanning speed was set to 20 mm/s.

Subsequently, derivatization was performed by immersing the plates in specific reagents for each chemical class using the Chromatogram Immersion Device III (Camag, Muttenz, Switzerland). Natural product reagent was used for flavonoids, anisaldehyde for polyphenols, and ninhydrin for amino acids, and no solvent was utilized for alkaloids. Prior to documentation under UV light at 366 nm and white light in reflectance mode, the plates were dried on a heating plate at 120 °C for 5 min.

### 3.4. NMR Analysis

Three independent samples were collected for each tea grade and extracted following the Bligh–Dyer protocol and then analyzed by NMR spectroscopy according to Patriarca et al. [27]. The hydrophilic phase was resuspended in 0.7 mL of D_2_O (Sigma-Aldrich, Milano, Italy) containing 3-(trimethylsilyl)-propionic-2,2,3,3-d4 acid sodium salt (TSP, 2 mM) (Sigma-Aldrich, Milano, Italy) as an internal chemical shift and concentration standard while the organic phase was resuspended in 0.7 mL of CDCl_3_ (Sigma-Aldrich, Milano, Italy) containing hexamethyldisiloxane (HMDSO, 2 mM) (Sigma-Aldrich, Milano, Italy) as an internal chemical shift and concentration standard.

The NMR experiments were carried out at 298 K on a JNM-ECZ 600R spectrometer (JEOL Ltd., Tokyo, Japan) operating at the proton frequency of 600 MHz and equipped with a multinuclear z-gradient inverse probe head. The monodimensional ^1^H NMR experiments were carried out for quantitative analysis, employing a presaturation pulse sequence for water suppression with a time length of 2 s (hydroalcoholic samples only), a spectral width of 9.03 KHz, and 64 k data points, corresponding to an acquisition time of 5.81 s. The pulse length of the 90° flip angle was set to 8.3 μs, the recycle delay was set to 5.72 s.

Bidimensional ^1^H-^1^H TOCSY and ^1^H-^13^C HSQC experiments were acquired for the resonance assignment. Quantities were expressed in mg/100 g through comparison of the relative integrals with the reference concentration and normalized to the number of protons (TSP: 9 protons) and to the starting fresh weight of the sample.

MATLAB^®^ R2023a (MathWorks, Natick, MA, USA) with the Statistics and Machine Learning Toolbox package was used as the program for univariate and multi-variate analysis of NMR samples, with a home-built script.

Univariate analysis was performed by one-way ANOVA test. Prior to this, normality and homoscedasticity of each metabolite were evaluated using, respectively, the Shapiro–Wilk and Brown–Forsythe tests [46]. ANOVA test was carried out on metabolites respecting those conditions, while non-parametric Kruskal–Wallis was carried out on the others. For multiple groups comparison, Bonferroni’s correction was applied to determine which categories were discriminated by these metabolites (*p* < 0.05) [47,48].

## 4. Conclusions

This study presents a comprehensive comparative analysis of three matcha tea grades (G1, G4, and FG) using a multi-analytical approach that integrates HPTLC and NMR spectroscopy. The findings highlight the strong influence of leaf maturity and processing techniques on the metabolite composition of matcha tea, which in turn reflects the different quality grades.

Amino acids, particularly theanine—a γ-amino acid with recognized health benefits—were most abundant in the ceremonial grade (G1) and progressively declined in G4 and FG samples. A similar decreasing trend was observed for caffeine and several organic acids, aligning with the understanding that younger tea leaves contain higher concentrations of nitrogen-rich and defensive metabolites.

Conversely, polyphenols, especially flavonoids such as kaempferol and rutin, were more concentrated in senescent leaves, which correspond to lower-grade samples. This inverse trend suggests a metabolic shift during leaf aging and post-harvest handling, where biosynthetic activity favors phenolic compounds over amino acid accumulation.

The combined use of HPTLC and NMR provided complementary insights into both targeted and untargeted metabolite profiling, offering a robust analytical framework for quality assessment. These results underscore the value of metabolomics in tea grading and authenticate the biochemical rationale behind matcha’s price and health claims. Ultimately, this work contributes to the field of functional foods by linking compositional profiles to grade-specific attributes, paving the way for enhanced consumer understanding and product standardization.

One potential limitation of this study is that it focused on three different matcha grades using extraction replicates, but not production replicates—i.e., samples from different batches or producers were not included. However, this work represents a preliminary investigation, and one of its main objectives was to identify marker compounds that could serve as reference indicators of quality. The long-term goal is to apply this approach to a broader set of samples from various batches and producers, preferably using HPTLC, which has proven to be not only effective but also faster and more cost-efficient. This does not exclude the use of NMR entirely, but rather envisions a more targeted application focused on the detection of specific characteristic markers, instead of the comprehensive untargeted metabolomics approach adopted in this initial phase. In this context, having assessed the compatibility and complementarity of the two analytical techniques may be valuable not only for the evaluation of different matcha tea grades, but also for broader applications to other matrices—such as botanical plant materials, nutraceuticals, foods, or other related products.

## Figures and Tables

**Figure 1 plants-14-01631-f001:**
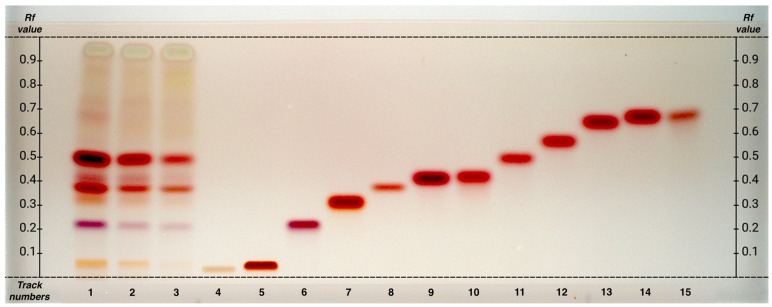
HPTLC fingerprints of amino acids from hydroalcoholic phase of matcha tea samples, visualized under white light after derivatization with ninhydrin reagents. Mobile phase: 1-butanol/acetone/acetic acid/water (7:7:2:4, *v/v*). Track assignments: 1 hydroalcoholic phase of G1, 2 hydroalcoholic extract of G4, 3 hydroalcoholic extract of FG, 4 cysteine, 5 lysine, 6 aspartic acid, 7 glutamine, 8 glutamic acid, 9 alanine, 10 threonine, 11 theanine, 12 valine, 13 isoleucine, 14 leucine, 15 tyrosine. For a clearer interpretation of the HPTLC plate, the y-axis shows the R*f* values and the x-axis the track numbers.

**Figure 2 plants-14-01631-f002:**
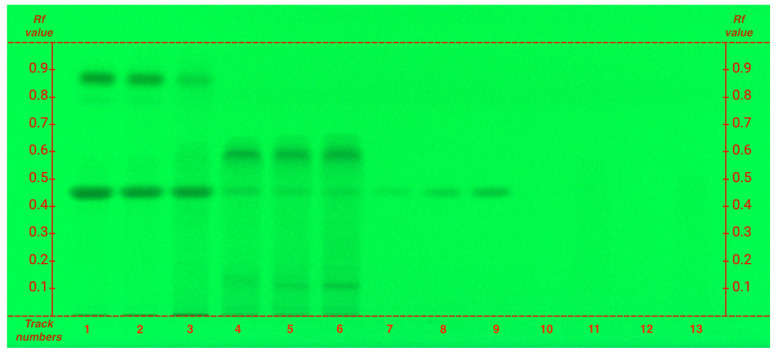
HPTLC fingerprints of alkaloids from both organic and hydroalcoholic phases of matcha tea samples, visualized under UV 254 nm without derivatization. Mobile phase: ethyl acetate/methanol/water (20:2.7:2 *v/v*). Track assignments: 1 organic phase of G1, 2 organic phase of G4, 3 organic phase of FG, 4 hydroalcoholic phase of G1, 5 hydroalcoholic phase of G4, 6 hydroalcoholic phase of FG, 7–9 caffeine (0,5; 1; 2 μg), 10 epicatechin, 11 epicatechin gallate, 12 epigallocatechin, 13 epigallocatechin gallate. Compounds 10–13 are visible only after derivatization with anisaldehyde. For clearer interpretation of the HPTLC plate, the y-axis shows the Rf values and the x-axis the track numbers.

**Figure 3 plants-14-01631-f003:**
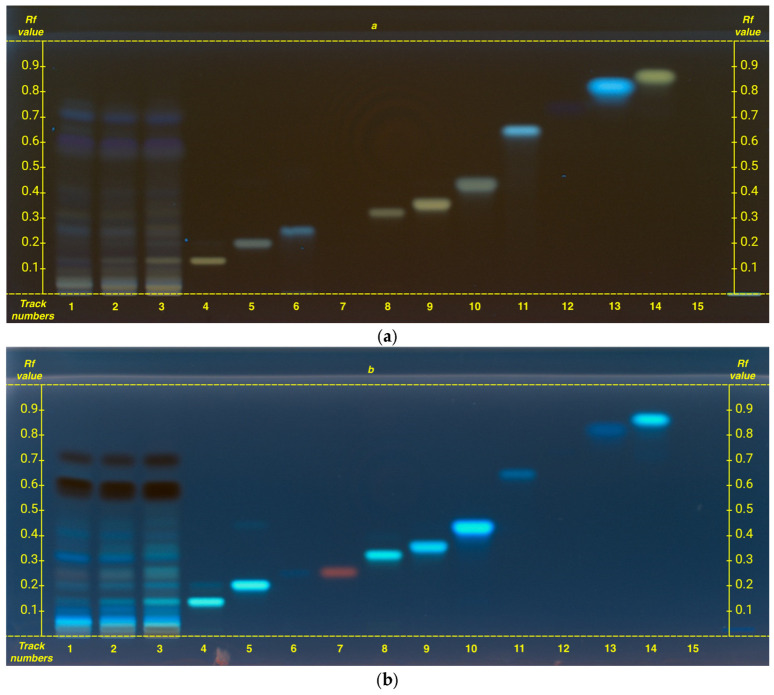
HPTLC fingerprints of flavonoids and organic acids from hydroalcoholic phase of matcha tea samples, visualized under UV 366 nm (**a**) after derivatization with natural product reagent and (**b**) after derivatization with anisaldehyde. Mobile phase: ethyl acetate/dichloromethane/acetic acid/formic acid/water (100:25:10:10:11, *v/v*). Track assignments: 1 hydroalcoholic phase of G1, 2 hydroalcoholic phase of G4, 3 hydroalcoholic phase of FG, 4 rutin, 5 kaempferol, 6 chlorogenic acid, 7 shikimic acid, 8 hyperoside, 9 luteolin 7-*O*-glucoside, 10 apigenin, 11 3,5-di-caffeoylquinic acid, 12 gallic acid, 13 caffeic acid, 14 quercetin, 15 cinnamic acid. Gallic acid appears in (**a**) as a very dark blue spot, while cinnamic acid is visible at 254 nm. For clearer interpretation of the HPTLC plate, the y-axis shows the Rf values and the x-axis the track numbers.

**Figure 4 plants-14-01631-f004:**
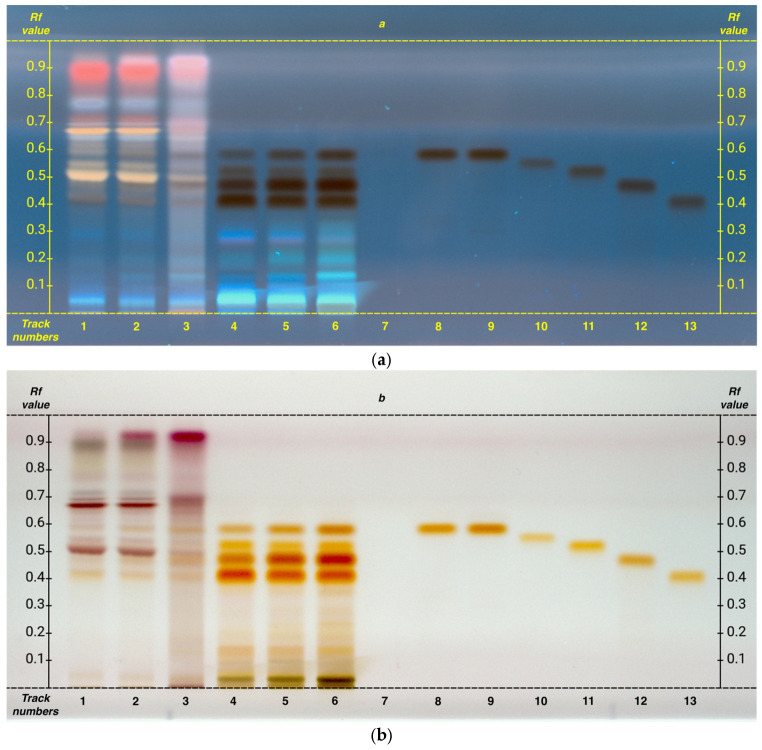
HPTLC fingerprints of selected polyphenols from both organic and hydroalcoholic phases of matcha tea samples, visualized after derivatization with anisaldehyde under UV 366 nm (**a**) and under white light (**b**). Mobile phase: toluene/acetone/formic acid (9: 9: 2, *v*/*v*). Track assignments: 1 organic phase of G1, 2 organic phase of G4, 3 organic phase of FG, 4 hydroalcoholic phase of G1, 5 hydroalcoholic phase of G4, 6 hydroalcoholic phase of FG, 7 caffeine, 8 catechin, 9 epicatechin, 10 catechin gallate, 11 epicatechin gallate, 12 epigallocatechin, 13 epigallocatechin gallate. Caffeine is visible only at 254 nm before derivatization. For clearer interpretation of the HPTLC plate, the y-axis shows the Rf values and the x-axis the track numbers.

**Table 1 plants-14-01631-t001:** The amino acid contents identified by HPTLC from the hydroalcoholic phase of matcha tea samples.

Amino Acid	G1	G4	FG	R*f*
Ala ^1^	+++	++	+	0.42
Arg	+++	++	+	0.09
Asn	−	−	−	0.34
Asp	+++	++	+	0.23
Cys	−	−	−	0.03
Gln	++	+	+	0.31
Glu	+++	++	+	0.38
Gly	−	-	−	0.30
His	−	-	−	0.03
Ile ^2^	+++	++	+	0.68
Leu ^2^	+++	++	+	0.69
Lys	++	+	+	0.07
Met	−	−	−	0.66
Phe	−	−	−	0.69
Pro	−	−	−	0.38
Ser	−	−	−	0.34
Thea	+++	++	+	0.51
Thr ^1^	+	+	+	0.43
Trp	−	−	−	0.72
Tyr	−	−	−	0.68
Val	−	−	−	0.58

The quantitative data indicated with a “+” sign are based on the comparison of the relative concentrations of the same metabolite across the different samples (G1, G4, and FG), as inferred from signal intensity. Specifically, “+” indicates low concentration, “++” medium concentration, “+++” high concentration, while “−” indicates not detected. ^1^ Alanine and threonine exhibit very similar R*f* values; however, preliminary analyses made it possible to identify the presence of alanine but not threonine. ^2^ Isoleucine, leucine, and tyrosine exhibit an *Rf* value very similar; however, tyrosine was not detected in the preliminary analysis.

**Table 2 plants-14-01631-t002:** Quantification and univariate test of the amino acids from the ^1^H NMR spectrum of hydroalcoholic extract of matcha tea.

mg/100 g	G1	G4	FG
Alanine	38.96 ± 1.82 a	44.25 ± 3.74 a	8.55 ± 0.14 b
Dimethylglycine	61.91 ± 4.37 a	68.73 ± 2.72 a	39.94 ± 3.22 b
Glutamine	765.24 ± 26.14 a	649.99 ± 68.04 b	301.45 ± 0.40 c
Isoleucine	9.25 ± 2.17 a	5.7 ± 0.08 b	5.06 ± 0.15 b
Leucine	6.54 ± 1.14 a	3.5 ± 0.21 b	1.58 ± 0.17 c
Lysine	102.37 ± 5.42 a	72.76 ± 7.97 b	41.81 ± 2.34 c
Theanine	978.42 ± 34.74 a	849.47 ± 71.05 a,b	351.85 ± 5.07 b
Threonine	17.22 ± 2.65 a	19.27 ± 2.12 a	6.55 ± 0.14 b
Valine	7.99 ± 3.90 a	5.98 ± 0.65 a,b	1.25 ± 0.57 b

The significant variables of the ANOVA test were indicated with “a”, “b”, “c” for the significative differences in one-way ANOVA with *p* < 0.05.

**Table 3 plants-14-01631-t003:** The flavonoid and organic acid content identified by HPTLC from the hydroalcoholic phase of matcha tea samples.

Flavonoid	G1	G4	FG	R*f*
Apigenin	−	−	−	0.43
Hyperoside	−	−	−	0.32
Kaempferol	+	++	+++	0.20
Luteolin	−	−	−	0.80
Luteolin 7-*O*-glucoside	−	−	−	0.84
Quercetin	−	−	−	0.86
Rutin	+	++	+++	0.14
**Organic Acid**	**G1**	**G4**	**FG**	**R*f***
3,5-Di-caffeoylquinic acid	−	−	−	0.65
Caffeic acid	−	−	−	0.82
Chlorogenic acid	+++	++	+	0.25
Cinnamic acid	−	−	−	0.92
Gallic acid	−	−	−	0.73
Protocatechuic acid	−	−	−	0.79
Shikimic acid	+++	++	+	0.26

The quantitative data indicated with a “+” sign are based on the comparison of the relative concentrations of the same metabolite across the different samples (G1, G4, and FG), as inferred from signal intensity. Specifically, “+” indicates low concentration, “++” medium concentration, “+++” high concentra-tion, while “−” indicates not detected.

**Table 4 plants-14-01631-t004:** Quantification and univariate test of the phenols from the ^1^H NMR spectrum of hydroalcoholic extract of matcha tea.

mg/100 g	G1	G4	FG
4-Hydroxybenzoic acid	20.39 ± 0.80 a	22.14 ± 0.52 b	8.37 ± 0.29 c
Cinnamic acid quinic ester	38.06 ± 3.89 a	96.43 ± 0.92 b	121.12 ± 7.12 c
Gallic acid quinic ester	96.91 ± 4.96 a	61.36 ± 4.66 b	26.42 ± 1.56 c
Protocatechuic acid	22.58 ± 12.53 a	Not detected	Not detected
Quinic acid	591.42 ± 40.14 a	506.86 ± 17.28 b	209.74 ± 17.26 c
Total chlorogenic acids	114.6 ± 18.03 a	139.55 ± 9.69 a	229.55 ± 32.03 b

The significant variables of the ANOVA test were indicated with “a”, “b”, “c” for the significative differences in one-way ANOVA with *p* < 0.05.

**Table 5 plants-14-01631-t005:** Quantification and univariate test the polyphenols from the ^1^H NMR spectrum of hydroalcoholic extract of matcha tea.

mg/100 g	G1	G4	FG
EC	217 ± 17.25 a	323.13 ± 5.55 b	257.29 ± 5.32 c
ECG	882.08 ± 52.97 a	780.18 ± 70.74 a	504.75 ± 125.83 b
EGC	1197.49 ± 36.51 a	2502.16 ± 142.64 b	2368.52 ± 21.19 b
EGCG	2664.99 ± 87.65 a	2521.54 ± 146.00 a	1706.27 ± 28.89 b

The significant variables of the ANOVA test were indicated with “a”, “b”, “c” for the significative differences in one-way ANOVA with *p* < 0.05.

**Table 6 plants-14-01631-t006:** Quantification and univariate test of each metabolite extracted from the ^1^H NMR spectrum of matcha tea.

mg/100 g	G1	G4	FG
Polyols and carbohydrates (hydroalcoholic extract)			
Fructose	334.55 ± 41.83 a	301.49 ± 36.42 a	125.53 ± 6.99 b
Fucose	2.78 ± 1.01 a	3.46 ± 1.24 a	Not detected
Myo-inositol	418.57 ± 44.37 a	415.16 ± 59.43 a	192.06 ± 6.85 b
Trehalose	755.68 ± 33.11 a	634.24 ± 40.28 b	265.79 ± 2.21 c
Sucrose	1313.45 ± 73.63 a	2502.57 ± 144.52 a,b	3262.88 ± 47.10 b
**Lipids and sterols (organic extract)**			
Cholestanol	0.9 ± 0.08 a	0.62 ± 0.14 b	Not detected
Ergostenol and ergosterol	33.18 ± 1.99 a	26.55 ± 1.65 b	19.66 ± 0.41 c
FA-omega 3	248.54 ± 10.37 a	198.67 ± 15.89 a,b	33.03 ± 0.94 b
FA-omega 6	54.06 ± 3.63 a	38.6 ± 4.75 b	10.41 ± 0.70 c
FA-omega 9	215.86 ± 9.30 a	148.72 ± 27.65 b	300.89 ± 8.19 c
Glycerophospholipids	13.68 ± 2.02 a	2.83 ± 3.46 b	Not detected
Triacylglycerols	18.83 ± 0.46 a	17.27 ± 2.54 a	22.85 ± 0.80 b
**Other metabolites (hydroalcoholic and organic extracts)**			
Betaine	36.68 ± 2.90 a	46.69 ± 14.62 a	26.06 ± 1.94 a
Choline	28.27 ± 1.47 a	19.53 ± 0.89 b	4.92 ± 0.18 c
α- and β-farnesene	13.16 ± 1.27 a	8.93 ± 0.17 b	0.73 ± 0.19 c
Methylguanidine	405.92 ± 18.30 a	416.41 ± 51.29 a	289.11 ± 2.61 b
Pheophytin A	16.49 ± 1.65 a	11.61 ± 1.22 b	7.53 ± 0.34 c
Pheophytin B	6.27 ± 0.36 a	4.62 ± 0.49 b	0.94 ± 0.27 c
Pyropheophorbide A	46.53 ± 2.28 a	28.56 ± 2.90 b	3.69 ± 0.21 c
Pyropheophorbide B	13.53 ± 2.35 a,b	9.87 ± 0.79 b	0.32 ± 0.11 c
**Unknown compounds (hydroalcoholic extract)**			
U01 (Myo inositol glycoside)	59.02 ± 4.55 a	278.08 ± 8.40 b	427.43 ± 3.29 c
U02 (Myo inositol glycoside)	72.62 ± 7.98 a	209.77 ± 10.99 b	156.37 ± 6.90 c
U03 (Caffeoyl quinic acid)	329.45 ± 23.31 a	394.96 ± 64.33 a	307.9 ± 27.13 a
U04 (uridine)	95.44 ± 5.29 a	65.1 ± 0.77 b	39.29 ± 1.50 c

The significant variables of the ANOVA test were indicated with “a”, “b”, “c” for the significative differences in one-way ANOVA with *p* < 0.05.

## Data Availability

The original contributions presented in this study are included in the article/Appendix A. Further inquiries can be directed to the corresponding author(s).

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
