# Peer review of "A Comparative Multianalytical Approach to the Characterization of Different Grades of Matcha Tea (Camellia sinensis (L.) Kuntze)"

_plants, 2025, doi:10.3390/plants14111631_

Round 1

Reviewer 1 Report

Comments and Suggestions for Authors

In their paper, the authors aimed to compare the metabolite composition of three different quality tea samples. To determine the amounts of compounds in the tea samples, NMR methodology was used. The results of NMR analyses were supported by the HPTLC technique. These results are interesting; however, further analyses are necessary to confirm the differences in metabolite composition among the different Matcha tea varieties. According to the “Materials and Methods” section, the study included only one sample each of grade 1 (G1), grade 4 (G4), and food grade (FG) Matcha. Three parallel extracts were prepared from each sample (i.e., three extracts from G1, three from G4, and three from FG). In this context, the parallels represent extraction replicates, and the results primarily support the reliability of the methodology. To robustly confirm the differences among the G1, G4, and FG samples, at least two additional samples of each grade, sourced from different manufacturers or representing different production batches, should be analyzed. Without these additional measurements, and based solely on the currently available data, the author should include an additional paragraph in the Conclusions section to acknowledge the limitations of their study.

Furthermore, some corrections/improvements need to be made as follows:

1) The type of extract used (hydroalcoholic or organic) should be consistently specified in the titles of all tables and figure captions. While some already include this information, others do not, and this inconsistency should be addressed.

2) According to Tables 3 and 4, shikimic acid was not detected in the extracts. However, in the text (page 10, lines 367–379), the authors discuss the shikimic acid content across samples. This apparent contradiction should be clarified and explained.

3) In the Title: “different grades” should be capitalized.

In conclusion, the manuscript has potential for publication but requires major revisions as outlined above.

Author Response

Thank you for your thoughtful and constructive comments. Please find attached a file with my detailed responses to each point raised. My responses are highlighted in red for clarity.

Reviewer 2 Report

Comments and Suggestions for Authors

The authors presented a complete study of HPTLC and NMR research on the detailed chemical composition of the Matcha product, which has recently become very popular as a substitute for tea beverages for the first time. I do not raise any critical remarks here. In particular, the analyses using the HPTLC planar chromatography technique in quantitative determinations are performed perfectly. My main serious remarks concern only a few issues, and they are as follows: 
1. The authors did not provide the batch numbers of the purchased product and who initially identified this product. 
2. I noticed a lack of analysis on the extract or water infusion from the tea powder, which would enhance both cognitive and consumer value, as this is the typical preparation method for this product. Since the authors suggested using hot water to prepare the beverage, discussing biocompatible extracts would be more relevant. Furthermore, in the context of alcohol extraction, the assessment of protein and amino acid composition seems inconsistent with the primary goals of the research. I would appreciate an explanation of this relationship. 
3. Finally, based on the literature data, I ask the authors to explain the safety aspects of using this product in the context of alkaloid derivatives and chlorogenic acid.

Author Response

(The authors gave the same response as above.)

Reviewer 3 Report

Comments and Suggestions for Authors

Dear authors, the information presented is very relevant and interesting, however, I believe it is necessary to integrate information that will contribute to your manuscript.

Introduction section

1.- It is recommended to include citations

bibliographic citations that describe the information contained in the lines (35-36) “Various types of tea are classified based on the degree of oxidation and the processing methods applied to the leaves”.

2.- It is suggested to include bibliographic citations that support the information presented in the following lines (50-53).

3.- It is recommended to include more information about the bioactive compounds produced by the type of matcha preparation as described in lines (52-53) “Thus, matcha consumption facilitates the ingestion of the entire spectrum of bioactive compounds found in tea leaves, not just those extracted in a traditional infusion”.

 4.- It is recommended to include the corresponding quotes in the information described in lines (55-62).

 5.- It is suggested to include information about which nutrients are generated by matcha shading, information described in line 59 “Shading plays a crucial role in the flavor and nutrient content of matcha”.

 6.- It is suggested to integrate bibliographic citation of the information presented in lines (75- 78) "Ceremonial grade matcha is reserved for traditional tea ceremonies due to its superior quality, with grade 1 representing the pinnacle of craftsmanship and refinement. Conversely, lower grade matcha, while less delicate, is ideal for recipes, which have been increasingly gaining recognition in recent times, such as ice creams, cookies, and smoothies, where its bolder and richer flavor can complement other ingredients.2

 7.- It is suggested that the objective of the research be more clearly stated.

 Materials and methods section

It is recommended to include the names of the chemical reagents in a specific way and the brands of each one of them in section 4.1. Chemicals and reagents

Results

2.1. Amino acids analysis

 The information presented in the results and their discussion in this section is clear, however, I consider that it is necessary to modify the formats of the figures and tables presented.

1.- It is suggested to include information on the name of the x-axis and y-axis of figure 1.

It is suggested to include in the information of the name of the table 1 the type of fraction presented.

3.- It is suggested to homogenize the number of decimals in the results presented in table 2.

2.2. Caffeine analysis

1.- It is suggested to include information in figure 2, it would be important to include the names of the x-axis and the y-axis.

2.3. Organic acids and phenols

2.- In the line it is described “Among the flavonoids, the presence of rutin and kaempferol was identified, with a 315 concentration increasing from G1 to FG”, however the results do not correspond when observing image 3 or the results presented in table 3, on the contrary the results presented indicate the opposite, and this result is related to what is described in line 316-318, could revise this information.

3., It is suggested to review the order of the presentation of the results of this section, because they are not described in a very clear way, for example, it is only indicated in line 333 “Flavonoids and organic acids contents are compared in Table 3”, however, these results are not discussed or presented in greater depth. 

4.- It is also suggested to format Figure 3, indicating the name corresponding to the “x” and “y” axes.

5.- It is suggested to format figure 4, indicating the name corresponding to the “x” and “y” axes.

Author Response

(The authors gave the same response as above.)

Round 2

Reviewer 1 Report

Comments and Suggestions for Authors

The authors improved their manuscript; it can be accepted for publication.

Author Response

Thank you 

Reviewer 2 Report

Comments and Suggestions for Authors

The authors addressed my comments and explained my concerns in their manuscript. In this revised form, the manuscript should be accepted for publication.

Author Response

Thank you 

Reviewer 3 Report

Comments and Suggestions for Authors

Dear authors, I thank you for having considered the observations made to your manuscript, I only consider important that the figures should still be modified in relation to the legends of the “x” and “y” axes of the graphs, it is suggested that the legends be integrated as part of the graph and not as part of the figure caption.  Likewise, the format of figure 3 is considered important, in this case, to integrate the figure captions a and b in the upper part of each of the graphs presented.

On the other hand, it is suggested that the statistical model described in each table should not be part of the title of the table, instead, the position should be changed to the lower part of the table.

Author Response

Thank you for your comment. We have modified the figures as requested. Previously, we had not made these changes because the images do not represent actual graphs but rather a chromatographic plate, and we did not consider axis legends essential in this context. However, we appreciate your suggestion and have now adjusted the figures accordingly. Additionally, the statistical models have been moved to the bottom of each table, as suggested.